

# Primary and secondary organic aerosols in 2016 summer of Beijing

Rongzhi Tang[1], Zepeng Wu[1], Xiao Li[1], Yujue Wang[1], Dongjie Shang[1], Yao Xiao[1], Mengren Li[1], Limin Zeng[1], Zhijun Wu[1], Mattias Hallquist[2], Min Hu[1], Song Guo[1,*]

[1]*State Key Joint Laboratory of Environmental Simulation and Pollution Control, College of Environmental Sciences and Engineering, Peking University, Beijing, 100871, PR China*

[2] *Atmospheric Science, Department of Chemistry and Molecular Biology, University of Gothenburg, Sweden*

[*] Correspondence to: Song Guo, songguo@pku.edu.cn



## 1 **Abstract**

To improve the air quality, Beijing government has employed several air pollution
control measures since 2008 Olympics. In order to investigate the organic aerosol
sources after the implementation of these measures, ambient fine particulate matters
were collected at a regional site Changping (CP) and an urban site Peking University
Atmosphere Environment MonitoRing Station (PKUERS) during the "Photochemical
Smog in China" field Campaign in summer of 2016. A chemical mass balance (CMB)
modeling and the tracer yield method were used to apportion the primary and
secondary organic sources. Our results showed that the particle concentration
decreased significantly during the last a few years. The apportioned primary and
secondary sources explained $62.8 \pm 18.3\%$ and $80.9 \pm 27.2\%$ of the measured OC at
CP and PKUERS, respectively. Vehicular emissions served as the dominant sources.
Except gasoline engine emission, the contributions of all the other primary sources
decreased. Besides, the anthropogenic SOC, i.e. toluene SOC, also decreased,
implying that deducting primary emission can reduce anthropogenic SOA. Different
from the SOA from other regions in the world, where biogenic SOA was dominant,
anthropogenic SOA was the major contributor to SOA, implying that deducting
anthropogenic VOCs emissions is an efficient way to reduce SOA in Beijing. Back
trajectory cluster analysis results showed that high mass concentrations of OC were
observed when the air mass was from south. However, the contributions of different
primary organic sources were similar, suggesting the regional particle pollution. The
ozone concentration and temperature correlated well with the SOA concentration.
Different correlations between day and night samples suggested the different SOA
formation pathways. Significant enhancement of SOA with increasing particle water
content and acidity were observed in our study, suggesting the aqueous phase
acid-catalyzed reactions may be the important SOA formation mechanism in summer
of Beijing.



## 1. Introduction

Beijing is the capital and a major metropolis of China. With the rapid economic
growth and urbanization, Beijing is experiencing serious air pollution problems, and
became one of the hotspots of $PM_{2.5}$ (particular matters with size smaller than 2.5μm)
pollution in the world (Guo et al., 2014; Xiang et al., 2017; Tian et al., 2016). Due to
the frequent haze events in Beijing, Beijing government has taken a series of control
measures in recent years, especially after 2008 Olympics, which may greatly
influence the primary and secondary particle sources. Therefore, elucidating the
current contributions of primary particle sources as well as secondary particle sources
is of vital importance. It is also important to compare with the previous results to
evaluate the effectiveness of the control measures and shed light on the influence of
the primary source emission control on the secondary aerosol formation.
Several studies regarding to the source apportionment of fine particles in Beijing have
been conducted using multifarious methods during the last few years (Yu et al., 2013;
Gao et al., 2014; Zheng et al., 2016b; Tan et al., 2014; Wang et al., 2009; Guo et al.,
2013). Receptor model is a commonly used method to apportion the particle sources
(Zhang et al., 2017; Zhou et al., 2017; Zhang et al., 2013; Song et al., 2006; Zheng et
al., 2005). Elemental tracers were previously used to apportion particulate matter
sources (Yu et al., 2013; Gao et al., 2014; Zheng et al., 2016b). However, elemental
tracer-based method was unable to distinguish sources that mostly emit organic
compounds instead of specific elements such as diesel/gasoline engines. Among all
the apportionment methods, chemical mass balance (CMB) model was one of the
most commonly used methods to apportion the primary organic sources of fine
particulate matter (Zhang et al., 2017; Hu et al., 2015; Schauer et al., 1996). Organic
tracers have been successfully used in several studies which aimed to quantify the
main sources of Beijing (Liu et al., 2016; Guo et al., 2013; Wang et al., 2009). Wang
et al. assessed the source contributions of carbonaceous aerosol during 2005 to 2007



(Wang et al., 2009). Guo et al. (Guo et al., 2013) and Liu et al. (Liu et al., 2016)
apportioned the organic aerosol sources using CMB model in summer of 2008 and a
severe haze event in winter of 2013. Both studies found that vehicle emission and coal
combustion were the dominant primary sources of fine organic particles. Tracer-yield
method has been considered as a useful tool to semi-quantify SOA derived from
specific VOCs precursors (Guo et al., 2012; Zhu et al., 2017; Zhu et al., 2016; Tao et
al., 2017; Hu et al., 2008). However, only a few studies have estimated secondary
organic aerosol in Beijing. Yang et al. (Yang et al., 2016) estimated the biogenic SOC
to OC during CAREBEIJING-2007 field campaign, and found that the SOC
accounted for 3.1% of the measured OC. Guo et al. (Guo et al., 2012) illustrated the
SOA contributions in 2008, and found that secondary organic carbon could contribute
a great portion ($32.5 \pm 15.9\%$) to measured organic carbon at the urban site. Ding et al.
(Ding et al., 2014) used the tracer-yield method to investigate the SOA loading on a
national scale and found that SOA, especially anthropogenic SOA played great role in
major city clusters of China.
In this study, we quantified 144 kinds of particulate organic species, including
primary and secondary organic tracers, at a regional site and an urban site of Beijing.
A CMB modeling and the tracer yield method were used to apportion the primary and
secondary sources of the organic aerosols in the 2016 summer of Beijing. The results
were compared with the previous studies to evaluate the effectiveness of control
measures on primary as well as secondary organic aerosols. Moreover, source
apportionment results from different air mass origins according to the back trajectory
clustering analysis were shown to investigate the influences of air mass from different
directions on the fine organic particle sources. Influencing factors of SOA formation,
i.e. temperature, oxidant concentration, aerosol water content, as well as particle
acidity were also discussed in this study to improve our understanding of SOA
formation under polluted environment.



## 2. Experimental

### 2.1 Sampling and Chemical Analysis



The measurements were conducted simultaneously at an urban site Peking University
Atmosphere Environment MonitoRing Station (PKUERS, 39°59′21″ N, 116°18′25″ E)
and a regional site Changping (CP, 40°8′24″N, 116°6′36″ E) 40km north of PKUERS
site during "Photochemical Smog in China" campaign, from May 16[th] to June 5[th],
2016 (see Fig. S1). The PKUERS site is set on the roof at an academic building on the
campus of Peking University in the northwest of Beijing. CP site is located on the
fourth floor of a building on the Peking University Changping campus of Changping.
Four-channel samplers (TH-16A, Tianhong, China) consisting of three quartz filter
channel and one Teflon filter channel, were employed to collect 12-h aerosol samples
at PKUERS and CP, respectively. The sampling flow rate was 16.7 L min$^{-1}$. Teflon
filters were weighed by a microbalance (Toledo AX105DR, USA) after a 24 h balance
in an environmental controlled room (temperature $20 \pm 1^{o}$C, relative humidity $40 \pm$
3%) for gravimetric analysis. Teflon-based samples were extracted by deionized water
to measure water-soluble inorganic compounds (WSICs), namely $Na^{+}$, $NH_4^{+}$, $K^{+}$,
$Mg^{2+}$, $Ca^{2+}$, $NO_3^{-}$, $SO_4^{2-}$ and $Cl^{-}$ by DIONEX ICS-2500 and ICS-2000
ion-chromatograph. One punch (1.45 cm$^{2}$) of quartz-based sample was then cut off to
analyze the EC and OC via thermal-optical method using Sunset Laboratory-based
instrument (NIOSH protocol, TOT). The other two quartz filters were then extracted
and analyzed for chemical composition and particulate organic matters. Some daytime
and nighttime samples were combined to ensure the detection of most organic
compounds. To better understand the chemical speciation, daytime samples were
separated from nighttime samples.
Authentic standards were used to identify and quantify the organic compounds. The
analytical methods used in this study referred to the previous work (Song et al., 2014).
Briefly, the samples were first spiked with a mixture of internal standard, including



ketopinic acid (KPA), 20 kinds of deuterated compounds, and one carbon isotope
$^{13}$C-substituted compound. The filters were then ultrasonically extracted with
methanol: dichloromethane (v:v=1:3) solvent in water bath (temperature < 30 °C) for
3 times. Each time was 20 min. The extracts were filtered, and then concentrated
using a rotary vacuum evaporator. An ultra-pure nitrogen flow was used to further
concentrate the extracts into 0.5-1 ml. Each extracted solution was divided into two
portions, one of which added BSTFA (BSTFA/TMCS = 99:1, Supelco) to convert
polar organic compounds into trimethylsilanized derivatives. Afterwards, the
derivatized and the untreated samples were analyzed by an Agilent 6890 GC-MS
System (MSD GC-5973N) equipped with an Agilent DB-5MS GC column (30 m ×
0.25 mm ×0.5 μm).

**2.2 Source Apportionment**

A chemical mass balance modelling developed by the U.S. Environmental Protection
Agency (EPA CMB version 8.2) was applied to determine the apportion of the
primary contribution of OC (Schauer et al., 1996). This receptor model solved a set of
linear equations using ambient concentrations and chemical source profiles. CMB
approach depends strongly on the representativeness of the source profile. In this
study, five primary source profiles including vegetative detritus (Rogge et al., 1993),
coal combustion (Zheng et al., 2005), gasoline engines (Lough et al., 2007), diesel
engines (Lough et al., 2007) as well as biomass burning (Sheesley et al., 2007) were
input into the model. Fitting species including EC,n-alkanes, levoglucosan,
17β(H)-21α(H)-norhopane,        17α(H)-21β(H)-hopane,        benzo(b)fluoranthene,
benzo(k)fluoranthene, benzo(e)pyrene, benzo(ghi)perylene, indeno(1,2,3-cd)pyrene.
The criteria for acceptable fitting results included the square regression coefficient of
the regression equation $R^2 > 0.85$ as well as the sum of square residual Chi-square
value $\chi^2 < 4$.





The tracer yield method was used to estimate the contributions of biogenic and
anthropogenic secondary organic aerosols using fixed tracers to SOC ratio ($f_{SOC}$)
based on laboratory experiments, which was 0.155 ±0.039 for isoprene, 0.231 ±0.111
for α-pinene, 0.0230 ± 0.0046 for β-caryophyllene and 0.0079 ± 0.0026 for toluene
(Kleindienst et al., 2007). The uncertainties of the estimation mainly derived from the
imparity in choosing the organic tracer compounds as well as the idealization of the
single-valued tracer mass fractions, which have been comprehensively discussed by
previous studies (Yttri et al., 2011;El Haddad et al., 2011) (Song et al., 2014). Despite
of its large uncertainties, the tracer-yield method was considered as a useful approach
up to now to roughly estimate the SOA contributions derived from individual
hydrocarbon precursors.

**3. Gaseous pollutants and particle chemical composition**

**3.1 Gaseous pollutants and meteorological conditions of the observation period**

Mixing ratios of gaseous pollutants and meteorological conditions during the
observation period were shown in Fig. S2 and Table S1. Compared with the results in
summer of 2010 (Zheng et al., 2016a), the gaseous mixing ratios $SO_2$ and CO were
lower than before owing to the desulfurization efforts made by the government.
Higher concentrations of NO and $NO_2$ were caused by the increasing number of
vehicles. The increment of ozone indicated the importance of secondary pollution.
Clearly, ozone concentration at CP was higher than that of PKUERS while other
pollutants were lower.
During the campaign, the average wind speed was low, showing average values of 2.3
± 1.4 m/s and 2.4 ±1.5 m/s at CP and PKUERS, respectively. The diurnal variations
of wind directions and speeds are exhibited in Fig. S2. The prevailing wind was from
south, with higher wind speed during the daytime.
To explore the influence of the air masses from different directions on fine particle
loading and sources, back trajectory analysis was performed using National Oceanic





and Atmospheric Administration (NOAA) Hybrid Single Particle Lagrangian
Integrated Trajectory (HYSPLIT) model. We calculated 36 h air mass back
trajectories arriving at two sampling site during the observation period using the
HYSPIT-4 model with a 1 °×1 ° latitude-longitude grid and the final meteorological
database. The model was run with the starting time of 0:00, 4:00, 8:00, 12:00, 16:00,
and 20:00 UTC). The arrival level was set at 200 m above ground level. The method
used in trajectory clustering was based on GIS-based software TrajStat
(http://www.meteothinker.com/TrajStatProduct.aspx). 36-h back trajectories staring at
200 m above ground level in CP and PKUERS were calculated every 4 hours during
the entire campaign and then clustered according to their similarity in spatial
distribution using the HYSPLIT4 software. Three-cluster solution was adopted as
shown in Fig. S3. The three clusters were defined as Far North West (Cluster 1, Far
NW), Near West North (Cluster2, Near WN), and South (Cluster 3). South cluster was
found to be the most frequent one, accounting for 52% at CP and 64% at PKUERS.
Clusters Far NW and Near NW accounted for 17% and 31%, 17% and 19% at CP and
PKUERS, respectively.

**3.2 Overview of PM$_{2.5}$ chemical composition**

In this study, daily PM$_{2.5}$ concentrations fluctuated dramatically from 6.7 μg m$^{-3}$ to
80.3 μg m$^{-3}$ at CP, and from 9.6 to 82.5 μg m$^{-3}$ at PKUERS, respectively. A paired
t-test was used to compare the mass concentrations at two sites. The results indicate
that the mass concentrations showed statistically non-significant difference,
suggesting the regional particle pollution in Beijing. PM$_{2.5}$ mass concentrations during
the summer of 2008 to 2016 in Beijing are summarized in Table 1. Guo et al. (Guo et
al., 2013) reported the average PM$_{2.5}$ concentrations during the summers of 2000 to
2008, which showed distinct decreasing tendency during 2000-2006 and then slightly
increased in 2007 due to unfavorable meteorological conditions. To better understand
the variation tendency of the PM$_{2.5}$ in the summer of Beijing, we compared the fine
particle matter data since 2008. Compared with 2008, the PM$_{2.5}$ concentrations


decreased from 92.3 ± 44.7 μg m$^{-3}$ to 88.2 μg m$^{-3}$ in 2009 and 62.7 μg m$^{-3}$ in 2010.
The mass concentration continued falling to 45.5 μg m$^{-3}$ in 2016. This decreasing is
attributed to the drastic emission control measures implemented by the Beijing
government since 2012. In spite of the prominent decrease of the PM$_{2.5}$ mass
concentrations, the aerosol loading in Beijing was still much higher than that in
developed countries (Tai et al., 2010; Barmpadimos et al., 2012; Park and Cho, 2011).
Fig. S4 showed the chemical composition of PM$_{2.5}$. In general, organic particulate
matters (OM, OC*1.6) and sulfate were the two dominant components, accounting for
more than 50% of the PM$_{2.5}$ mass concentration during the field campaign. The
average concentration of total WSICs for CP was 17.4 ± 11.5 μg m$^{-3}$, higher than that
of PKUERS (12.2 ± 8.5μg m$^{-3}$). Among the WSICs, secondary inorganic ions (sulfate,
nitrate, and ammonium) were the most abundant compounds, indicating secondary
particles played great roles in the summer of Beijing. The higher sulfate proportion
could be explained by the increased photochemical conversion of sulfur dioxide to
sulfate aerosol (Xiang et al., 2017).
Carbonaceous aerosols, i.e. organic carbon (OC) and elemental carbon (EC) were also
great contributors to PM$_{2.5}$ concentrations. Higher proportion of OC and EC at
PKUERS demonstrated severe carbonaceous pollution in urban Beijing, which might
have close correlation with the higher traffic flow, coal/wood combustion by residents
and industrial emissions (Wang et al., 2006; Dan et al., 2004; Cao et al., 2004).
Comparison of the OC, EC concentrations from 2008 to 2016 were also listed in Table
1. Unlike PM$_{2.5}$, OC concentration at PKUERS showed a higher OC concentration
(11.0 ± 3.7 μg m$^{-3}$) compared with that in 2008 (9.2 ± 3.3μg m$^{-3}$), suggesting organic
aerosol pollution becomes more and more important. EC concentration decreased
dramatically to 0.7 ± 0.5 μg m$^{-3}$ at CP and 1.8 ± 1.0 μg m$^{-3}$ at PKUERS, which
showed the lowest value since 2000. This could be attributed to the implementation of
air pollution prevention and control action plan enacted by the state council since

217    2013.



To evaluate the influences of the air masses from different directions on the $PM_{2.5}$
loadings during the campaign, three categories were divided according to the back
trajectory clustering analysis (See Fig. S5). In general, cluster south represented the
most polluted air mass origin followed by clusters Near WN and Far NW, which was
in accordance with previous studies demonstrating severe aerosol pollution in
southerly air flow in summer of Beijing (Huang et al., 2010; Sun et al., 2010).
**3.3 Concentration of particulate organic species from different air mass origins**
The organic species (except secondary organic tracers) were divided into 12
categories. Their concentrations in different directions according to the back trajectory
clustering were shown in Fig. S6. Detailed information for each class at the two sites
could be found in the supplementary material (Fig. S7). Cluster south showed higher
particulate organic matter concentration, followed by cluster near WN and far NW,
indicating more severe aerosol pollution from the south. Our result consists with the
previous studies that more pollution emissions are from the south area of Beijing than
those from the north (Wu et al., 2011; Zhang et al., 2009).
Dicarboxylic acid was the most abundant species among all the components,
demonstrating the great contribution of the secondary formation to the organic
aerosols in the summer of Beijing (Guo et al., 2010). A series of n-alkanes ranging
from C12 to C36 were analyzed. Their distribution during the observation period was
shown in Fig. S7 (a). The maximum-alkane concentration species ($C_{max}$) were C27
and C29. The odd carbon preference was an indicative of biogenic sources (vegetative
matters and biomass burning) (Huang et al., 2006; Rogge et al., 1993). In this study,
total PAHs were much lower than previous studies in summer of Beijing, suggesting
the effectiveness of the control strategies since 2013 (Wang et al., 2009). According to
Fig. S7 (c), five ring PAHs were dominant species among all the species, followed by
four-ring and six-ring PAHs. In total, four to six ring PAHs had higher abundancy,
accounting for more than 60% of the total PAHs. The result was much similar with





previous studies that the distribution of PAHs was impacted by the volatility of PAHs
and the temperature (Wang et al., 2009; Guo et al., 2013). Saccharide was considered
to originate from biomass burning (Simoneit et al., 1999). In this study, we quantified
three sugar compounds including levoglucosan, manosan and galactosan, in which
levoglucosan was considered as a good tracer for biomass burning. The average daily
mass concentration of levoglucosan at CP and PKUERS were $53.03 \pm 39.26$ ng m$^{-3}$
and $59.87 \pm 38.93$ ng m$^{-3}$, respectively. It's worth mentioning that the levoglucosan
concentration was the lowest in recent years (Cheng et al., 2013; Guo et al., 2013).
Hopanes have been considered as markers for oil combustion (Lambe et al., 2009),
vehicles (i.e. gasoline-powered and diesel-powered engine) (Cass, 1998; Lough et al.,
2007) and coal combustion(Oros and Simoneit, 2000). Nevertheless, contributions of
coal combustion to hopanes were much less than that of vehicle exhaustion.
Concentrations of quantified hopanes including 17α(H)-22,29,30-trishopane,
17β(H)-21α(H)-norhopane, and 17α(H)-21β(H)-hopane of CP and PKUERS are
shown in Fig. S7(d). The total average concentrations of hopanes were $3.05 \pm 1.53$ ng
m$^{-3}$ for CP and $3.90 \pm 2.06$ ng m$^{-3}$ for PKUERS. The hopanes concentrations at urban
site PKUERS were much higher than that of CP, which could probably explained by
the heavier vehicle emissions in the urban area. The concentrations of primary organic
tracers used in CMB model were listed in Table S2.

**3.4 Biogenic and anthropogenic SOA tracers**

Table S3 compared the SOA tracers measured in this work with those in other regions
in the world as well as that observed in Beijing 2008. The sites for comparison
include an urban background site at Indian Institute of Technology Bombay, Mumbai,
India (IITB) (Fu et al., 2016), an outflow region of Asian aerosols and precursors
Cape Hedo, Okinawa, Japan (CH) (Zhu et al., 2016), a residential site at Yuen Long,
Hong Kong (YL) (Hu et al., 2008), three industrial sites at Cleveland Ohio (CL, data
was averaged among the three sites), a suburban site in the Research Triangle Park



North California (RTP). The detailed information about these sites were summarized
in the supplementary material.
Three isoprene-SOA tracers i.e. two 2-methyltetrols (2-methyltheitol and
2-methylerythritol) and 2-methylglyceric acid were detected. The summed
concentration of the isoprene-SOA tracers ranged from 3.7 to 62.3 ng m$^{-3}$ at CP and
8.6 to 46.5 ng m$^{-3}$ at PKUERS. The concentration was higher than that of IITB and
CH. Compared with the isoprene-SOA tracers in 2008, the concentrations in 2016
were lower.
Nine α-pinene tracers were identified. The sum of the tracers ranged from 20.9 to
282.3 ng m$^{-3}$ at CP and 50.0 to 161.4 ng m$^{-3}$ at PKUERS, which had similar
distribution pattern with that measured in 2008 Beijing and YL. The total α-pinene
tracer concentrations were lower than those in 2008, while still much higher than the
concentrations in other regions of the world.
β-caryophyllinic acid is one of the oxidation products of β-caryophyllene which is
considered as a tracer for β-caryophyllene SOA. In this study, β-caryophyllinic acid
concentrations ranged from 1.4 to 16.7 ng m$^{-3}$ at CP, and 0.9 to 12.0 ng m$^{-3}$ at
PKUERS, with average daily average concentrations of 6.1 ±3.5 ng m$^{-3}$ and 6.0 ±2.8
ng m$^{-3}$ for CP and PKUERS, respectively. The values were lower than those at YL and
RPT, higher than that measured at Yufa and PKUERS in 2008.
2,3-Dihydroxy-4-oxopentanoic acid is deemed as a tracer for toluene SOA. Our
results showed that the 2,3-Dihydroxy-4-oxopentanoic acid concentration was 9.7 ±
7.3 ng m$^{-3}$ at CP and 11.0 ±3.7 ng m$^{-3}$ at PKUERS. Compared with other regions of
the world, the concentrations of 2,3-Dihydroxy-4-oxopentanoic acid was much higher,
implying higher contributions of anthropogenic sources at Beijing. However, the
concentrations in CP were lower than that of PKUERS.



## 4. Primary sources and secondary formation of organic aerosols

### 4.1 Contributions of primary and secondary organic aerosols

A CMB model and the tracer-yield method were used to quantify the contributions of primary and secondary sources to the ambient organic carbon (See Fig. 1). On average, the primary sources accounted for $42.6 \pm 15.4\%$ and $50.4 \pm 19.1\%$ of the measured OC at CP and PKUERS, respectively. The vehicle emissions were the dominant primary sources, with the contributions of $28.8 \pm 14.8\%$ and $37.6 \pm 19.3\%$ at PKUERS and CP, respectively, implying the urgency to control vehicular exhaustion in urban areas. Despite of the lower contribution of the gasoline exhaust at PKUERS, the mass concentration of the gasoline exhaust was higher compared with that of CP given the higher OC loading at PKUERS. The contributions of biomass burning were $3.9 \pm 2.6\%$ and $5.0 \pm 2.2\%$ at CP and PKUERS, respectively, showing the higher concentrations at night. The drastic change of the biomass burning contribution in CP at night was due to occasional burning activities at night. Coal combustion contributed $5.8 \pm 5.5\%$ and $4.6 \pm 2.6\%$ of the measured OC at CP and PKUERS. The higher contribution at CP was due to more burning activities in the suburban areas.

The secondary organic sources accounted for $20.2 \pm 6.7\%$ of the organic carbon at CP, with $1.6 \pm 0.4\%$ from isoprene, $4.4 \pm 1.5\%$ from α-pinene, $2.7 \pm 1.0\%$ from β-caryophyllene and $12.5 \pm 3.4\%$ from toluene. As for PKUERS, the secondary organic sources took up $30.5 \pm 12.0\%$ of the measured OC, in which isoprene was responsible for $2.3 \pm 0.9\%$, α-pinene for $5.6 \pm 1.9\%$, β-caryophyllene for $3.6 \pm 2.6\%$ and toluene for $19.0 \pm 8.2\%$. Haque et al. (Haque et al., 2016) used tracer-based method to apportion the organic carbon and results showed that the biogenic SOC was responsible for 21.3% of the total OC with isoprene SOC contributing 17.4%, α/β-pinene SOC contributing 2.5% and β-caryophyllene SOC contributing 1.4% in the summer of Alaska, implying the significant contributions of the biogenic SOA to the



loading of the organic aerosol. Our results exhibited that the biogenic SOA
concentration was comparable or even high than that at some forest sites in other
places of the world (Miyazaki et al., 2012; Stone et al., 2012; Claeys et al., 2004;
Kourtchev et al., 2008). The SOA formation mechanism is complicated. A possible
reason is the high oxidation capacity in China. More work is still needed to
investigate the SOA formation mechanism under Air Pollution Complex in China.
Stone et al. (Stone et al., 2009) discovered that primary and secondary sources
accounted for $83 \pm 8\%$ of the measured organic carbon, with primary sources
accounted for $37 \pm 2\%$ and SOC contributed for $46 \pm 6\%$ with $16 \pm 2\%$ from biogenic
gas-phase precursors and $30 \pm 4\%$ from toluene using CMB model and tracer-based
method at Cleveland with heavy industries, implying that anthropogenic sources
played great roles in the formation of SOA. Our results showed a similar with the
results published by Stone et al., where anthropogenic sources i.e. toluene derived
SOC dominated the apportioned SOC. Our research revealed an important point that
controlling SOA seems feasible in the developing countries like China. It is difficult
to control SOA in developed countries, since biogenic SOA are dominant. However,
deducting anthropogenic precursors may be an efficient way to reduce the SOA
pollution where anthropogenic SOA is dominant. On average, $62.8 \pm 18.3\%$ and $80.9$
$\pm 27.2\%$ of the measured OC were apportioned at CP and PKUERS, respectively.
About $36.3 \pm 18.1\%$ and $29.3 \pm 15.6\%$ of the OC sources remained unknown, which
were probably composed of uncharacterized primary or secondary sources. Further
research is needed to explain the unapportioned sources of OC.
Due to the drastic emission control measures taken by the Beijing government, the
primary and secondary sources in Beijing may change greatly. Fig. 2 displayed the
comparison of the sources between 2008 and 2016 at the same site PKUERS. We
compared the average contributions by percentage rather than the mass concentration.
In general, primary sources contributed $50.4 \pm 19.1\%$ of the measured OC in 2016,
closely correlated to the increasing contribution of vehicular emissions. Gasoline



engines accounted for 18% of the measured OC, showing an enhancement of 80% with respect to 2008. This might be related to the rising number of the vehicles in Beijing. In comparison, diesel exhaust decreased by 12.5% due to the strict control measures made by the government. A 28.5% and 20% reduction of coal combustion and biomass burning could also be found due to the drastic measures made by the government. Compared with 2008, contributions of secondary organic aerosol decreased by 29.4%. However, the contribution of toluene SOC was the highest among the apportioned SOC, which was different from the results of the most developed countries in the world. In summary, the contributions of most POA decreased in recent years, except for gasoline exhaust, indicating more efforts should be made to control the gasoline emission. The apportioned SOC was also decreased with toluene SOC served as the dominant source. Our results revealed that deducting anthropogenic precursors may be an efficient way to control SOA pollution in China.

**4.2 Organic aerosol sources from different air mass origins**

The regional sources and transport of air pollutants exert profound impacts on air quality of Beijing. To better understand the regional impacts on the primary and secondary aerosol sources of Beijing, source apportionment results when air mass from different origins were shown in Fig. 3. Vehicular emissions i.e. gasoline and diesel exhaust showed identical contributions from different air mass origins (31.0% from south vs 31.3% from Near WN vs 31.7% from Far NW) at PKUERS, demonstrating the vehicular pollution could mostly be attributed to the vehicular emission at the local site. However, the contribution of vehicular emission at CP showed significant difference from different air mass origins, with lowest contribution when air mass was from far northwest. This might be explained by regional transport from different directions. Comparable contributions of coal combustion and biomass burning were found at CP and PKUERS from different air mass origins, implying the regional pollution in Beijing. Similarly, biogenic SOC showed similar contributions from different air mass origins both at the regional site and the urban site. From all the



directions, the toluene SOC (anthropogenic source) was the largest contributor to
apportioned SOC, with higher concentrations at the urban site PKUERS. On the
whole, most of the sources showed comparable contribution from different air mass
origins, implying the pollution in Beijing was regional.

**4.3 Influencing factors for secondary organic aerosol formation in the summer of**

**Beijing**
Laboratory experiments have revealed that several factors can influence the SOA
formation, e.g. oxidants (OH radical, ozone etc.), temperature, humidity, particle
water content and acidity. In this work, the relationships between estimated SOA and
these factors were investigated to better understand the SOA formation in Beijing.
**SOA formation from ozonolysis**
Ozone is considered as an important oxidant for SOA formation. Fig. 4 (a)(b) showed
the correlation with ozone mixing ratio and SOC. It is clear that SOC increased
significantly with the increasing of ozone mixing ratio, which is consistent with
previous studies in Beijing (Guo et al. 2012). Different correlations were found
between day and night samples, with better correlation for the daytime samples at
both sites, implying SOA may have other formation mechanism at night other than
ozonolysis. At CP, the growth rate of SOC with $O_3$ was similar for day and night
samples, which was 0.02 $\mu g\ m^{-3}$ per ppbv ozone. For PKUERS, the increment rate of
the SOC towards ozone was 0.04 $\mu g\ m^{-3}$ and 0.02 $\mu g\ m^{-3}$ per ppbv ozone at day and
night, respectively.
**Influence of temperature and relative humidity on SOA formation**
Temperature was considered as a great influencing factor on SOA formation. On the
one hand, higher temperature promoted the evaporation of the semi volatile SOA. On
the other hand, high-temperature conditions would favor the oxidation, which would
accelerate the SOA formation (Saathoff et al., 2009). Fig. 4 (c) (d) showed the





variation of SOC concentrations with the temperature. In this study, SOC
concentration showed positive correlation with temperature at CP and PKUERS,
demonstrating that temperature favors the SOA formation in the summer of Beijing.
Moreover, different correlation of the day and the night samples imply the different
pathways of SOA formation. However, poor relations could be found between SOC
and RH.

**412    Effects of aqueous-phase acid catalyzed reactions on SOA formation**

Aerosol water and acidity have been considered to have great impact on the
aqueous-phase SOA formation (Cheng et al., 2016). To figure out the influences of
water content and aerosol acidity on the aqueous-phase reactions, ISORROPIA-II
thermodynamic equilibrium model was used (Surratt et al., 2007). The model was set
at forward mode, based on the concentrations of particle phase $Na^+$, $NH_4^+$, $K^+$, $Mg^{2+}$,
$Ca^{2+}$, $NO_3^-$, $SO_4^{2-}$, $Cl^-$ and gaseous $NH_3$ as well as ambient temperature and RH.
Results showed that the average aerosol water content at CP was 3.87 $\pm$ 3.73 $\mu g\ m^{-3}$,
higher than that at PKUERS (1.83 $\pm$ 1.81 $\mu g\ m^{-3}$). The water content was lower in
2016 than that in 2008. The estimated SOC concentration showed good correlations
with water content at both sites. Compared with CP, the correlation factor in PKUERS
was better, implying that aqueous phase reaction may be more important in the urban
area. Different correlation could be found at different liquid water contents, especially
for CP, where liquid water contented spanned a wide range, implying that different
mechanisms may exist at different liquid water content.
In this study, modeled $H^+$ concentration and SOC showed significant correlation
(p<0.05) at the two places, which indicated that acid-catalyzed reaction might provide
a crucial pathway for the SOA formation in the summer of Beijing. Laboratory studies
showed that acid-catalyzed reactive uptake might play great role in the enhancement
of SOA (Zhang et al., 2014; Surratt et al., 2010; Jang et al., 2002). However, contrary
conclusions were made by other group, demonstrating the inconsistence of the aerosol





acidity and the SOA formation (Wong et al., 2015; Kristensen et al., 2014). The
contradiction might give the facts that the impacts of the acidity on the SOA loading
varied from place to place, determined by the specific environmental conditions.
Linear regression showed that the enhancement of SOC with modeled $H^+$
concentration were at a value of 0.02 $\mu g\ m^{-3}$ per nmol $H^+$, which was lower than the
previous results (0.046 for PKUERS, and 0.041 for Yufa, 2008). Offenberg et al.
(Offenberg et al., 2009) discovered good correlation between SOC and $[H^+]_{air}$, with
$R^2$ value of 0.815. Besides, a one nmol $m^{-3}$ $[H^+]_{air}$ would give rise to 0.015 $\mu g\ m^{-3}$
SOC increase from the oxidation of α-pinene in the chamber experiment. In the
present work, different correlations could be found at different modeled $H^+$
concentrations where apportioned SOC increased significantly as the $H^+$
concentration increased and then increased slowly at a certain level, showing gradient
growth at different hydrogen-ion concentrations. Therefore, aqueous phase
acid-catalyzed reactions may influence the SOA formation through different
mechanisms at varied level of liquid water concentration and aerosol acidity.

## 5. Conclusion

High concentrations of fine particles were observed during the "Campaign on
Photochemical Smog in China", with the average mass concentrations of 45.48 ±
19.78 $\mu g\ m^{-3}$ and 42.99 ± 17.50 $\mu g\ m^{-3}$, at CP site and PKUERS site, respectively.
Compared with previous studies, the concentrations of $PM_{2.5}$, EC and estimated SOC
decreased significantly, due to the drastic measures implemented by the government
in the recent years. However, OC showed a higher concentration, suggesting
particulate organic matters become more and more important in Beijing. CMB
modeling and tracer-yield method were used to apportion the primary and secondary
organic aerosol sources. The apportioned primary and secondary OC accounted for
62.8 ± 18.3% and 80.9 ± 27.2% of the measured OC at CP and PKUERS, respectively.
Vehicle emissions i.e. diesel and gasoline engine emissions were the major primary
organic aerosol sources, which contributed to 28.8 ± 14.8% and 37.6 ± 19.3% of the





OC at CP and PKUERS, respectively. Compared with the results of the previous work,
the gasoline engine emission contributed almost twice of that in 2008 (18.0% vs
10.0%), while the contribution of diesel engine emission decreased by 12.5%
compared with the result in 2008. Besides, the contributions of biomass burning and
coal combustion both decreased. The apportioned biogenic and anthropogenic SOC
can explain $20.2 \pm 6.7\%$ and $30.5 \pm 12.0\%$ of the measured OC at CP and PKUERS,
respectively. The contribution of toluene SOC is the highest among the apportioned
SOC, which is different from the results of the most developed countries in the world.
Our results revealed an important point, which is that controlling SOA seems feasible
in the developing countries like China. It is difficult to control SOA in developed
countries, since biogenic SOA are dominant. However, deducting anthropogenic
precursors may be an efficient way to reduce the SOA pollution where anthropogenic
SOA is dominant. Back trajectory clustering analysis showed that the particle source
contributions were similar when air masses were from different directions, suggesting
the regional organic particle pollution in Beijing. However, the higher organic particle
loading from south cluster indicates that there were more emissions from southern
region of Beijing. The present work also implied that the aqueous phase
acid-catalyzed reactions may be an important SOA formation mechanism in summer
of Beijing.





**Acknowledgement**


481    This research is supported by the National Key R&D Program of China: Task

3 (2016YFC0202000), the National Natural Science Foundation of China (21677002),
framework research program on 'Photochemical smog in China" financed by Swedish
Research Council (639-2013-6917).



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

Contributions and source identification of biogenic and anthropogenic

hydrocarbons to secondary organic aerosols at Mt. Tai in 2014, Environmental

Pollution, 220, 863-872, 10.1016/j.envpol.2016.10.070, 2017.



**Table**

Table 1. Summer PM$_{2.5}$ mass concentrations in Beijing from 2008-2016, average ± standard deviation (μg m$^{-3}$).

| Year/Month | 2008/07 | 2009/07 | 2010/05 | 2016/05-06 | 2016/05-06 |
|---|---|---|---|---|---|
| Site | PKUERS | PKUERS | PKUERS | CP | PKUERS |
| | (μg m-3) | (μg m-3) | (μg m-3) | (μg m-3) | (μg m-3) |
| PM$_{2.5}$ | 92.3±44.7 | 88.2±52.3 | 62.7±36.5 | 43.0±17.5 | 45.5±19.8 |
| OC | 10.4±2.9 | 8.5±2.5 | 8.9±4.5 | 8.9±3.2 | 11.0±3.7 |
| EC | 3.3±1.5 | 2.5±0.9 | 2.1±1.1 | 0.7±0.5 | 1.8±1.0 |
| Ref. | (Guo et al., 2012) | (Zheng et al., 2016a) | (Zheng et al., 2016a) | This study | This study |



**Figure captions**

Fig. 1 Concentrations of organic carbon from primary and secondary organic sources
at (a) CP and (b) PKUERS as well as their contributions to the measured organic
carbon at (c) CP and (d) PKUERS (%).

Fig. 2 Comparison of the sources at PKUERS between 2016 and 2008

Fig. 3 Particle sources from different air mass origins

Fig. 4 Correlations between SOC and different influencing factors (a)-(b) ozone,
(c)-(d) temperature, (e)-(f) water and (g)-(h) $H^+$ concentratio





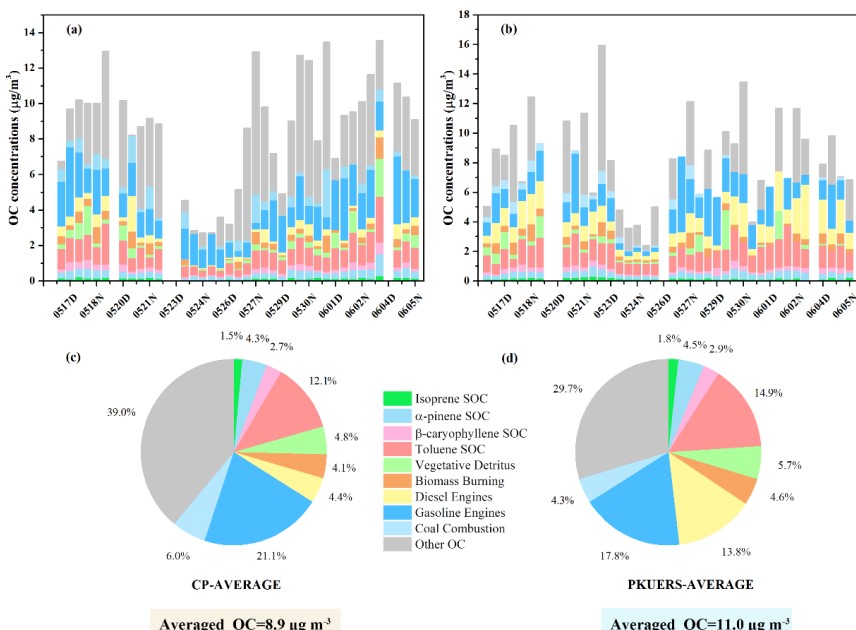

Fig. 1 Concentrations of organic carbon from primary and secondary organic sources at (a) CP and (b) PKUERS as well as their contributions to the measured organic carbon at (c) CP and (d) PKUERS (%).





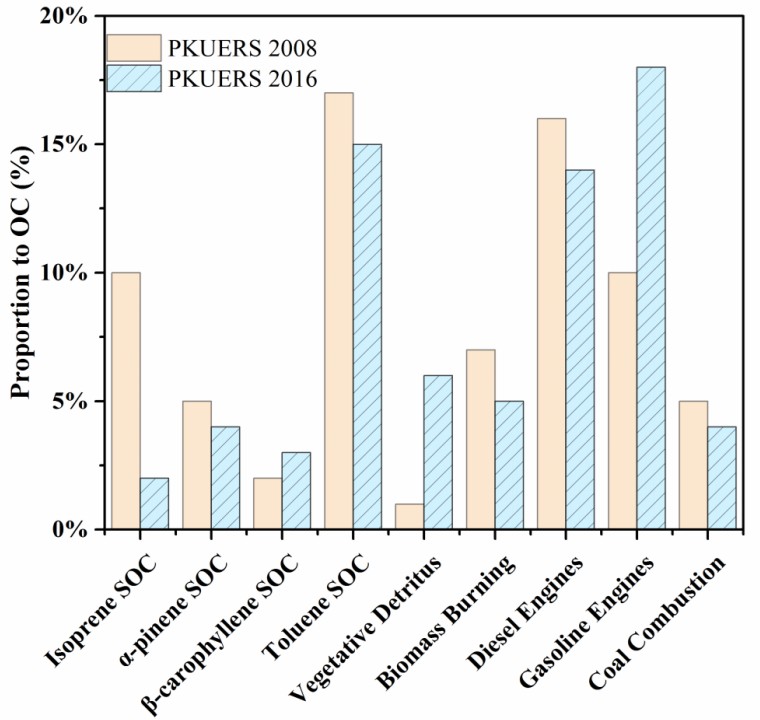

Fig.2 Comparison of the sources at PKUERS between 2016 and 2008 (Guo et al. 2012)



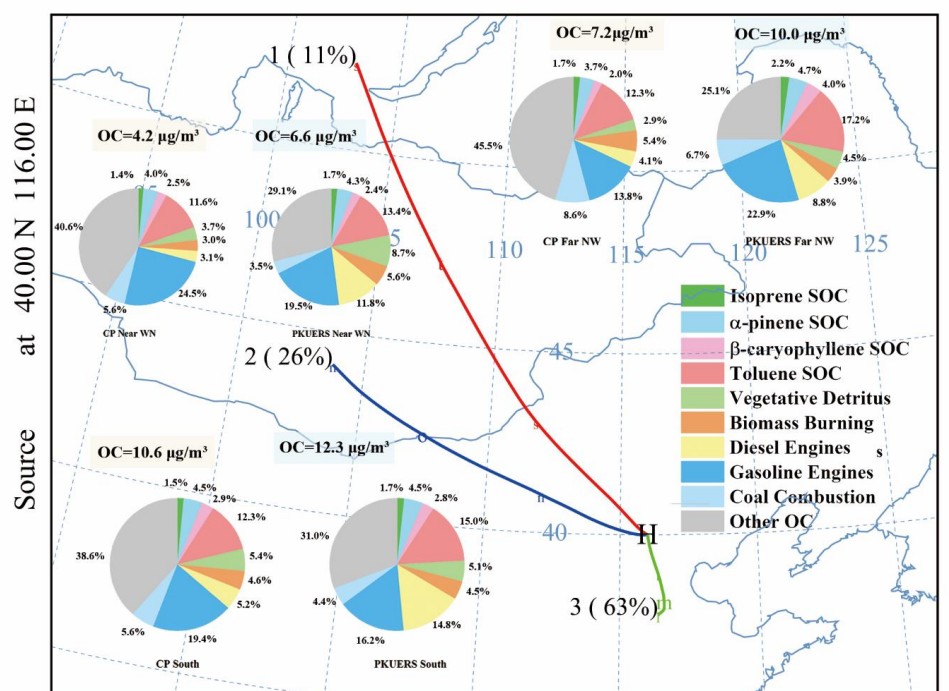

Fig. 3 Particle sources from different air mass origins





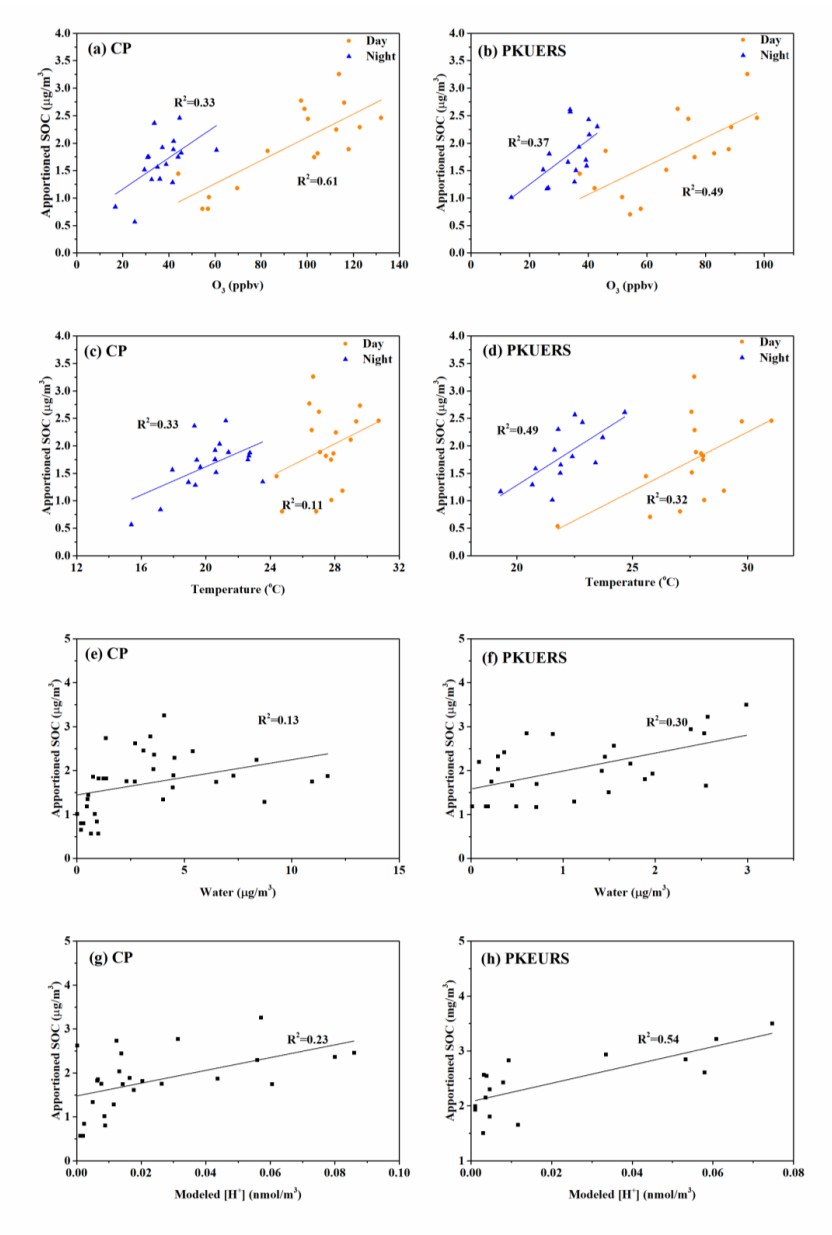

Fig. 4 Correlations between SOC and different influencing factors (a)-(b) ozone, (c)-(d) temperature, (e)-(f) water and (g)-(h) H$^+$ concentration