# Peer review of "Primary and secondary organic aerosols in 2016 summer of Beijing"

_Atmospheric Chemistry and Physics, 2017_

## Referee Comment (RC1) · Anonymous Referee #1 · 12 Dec 2017

Review of "Primary and secondary organic aerosols in 2016 summer of Beijing" by Tang et al.

This manuscript quantified 144 particulate organic species and applied chemical mass balance (CMB) model to investigate the sources of organic aerosol at two different sites (CP and PKUERS) in Beijing. The authors found that the primary sources accounted for 42.6% and 50.4% of the measured OC at CP and PKUERS, respectively, which are larger than the contribution from secondary sources. Among the secondary sources, anthropogenic VOCs contributes more to SOA than biogenic VOCs. By comparing with previous studies, the authors showed that the PM and EC concentrations have decreased. The OC concentration from many sources have decreased, with the exception of OC from gasoline engine emissions. This comparison sheds light on the evaluation of regulation policies. At last, the authors investigated the relationship between SOC concentration with temperature, ozone concentration, aerosol water content, and particle acidity. Overall, the manuscript is well-written and the data analysis is solid. I recommend publication after minor revisions to address the main comments below.

Major comments

1.     The uncertainty with tracer yield method should be better discussed. In that method, a laboratory-derived single-valued mass fraction from Kleindienst et al. (2007) is used. However, in the atmosphere, the mass fraction of molecular markers in SOA from a specific source is highly dependent on the oxidation conditions and the history of the airmass. How representative is the mass fraction value used in this study?

2.     I find it very intriguing that while PM and EC concentrations have decreased from 2008 to 2016, the OC concentration is relatively constant (Table 1). As shown in Fig. 2, the contributions from many sources to OC have decreased, with the exception of vegetative detritus and gasoline engines. By eyeballing, the increases in vegetative detritus and gasoline engines seems smaller than the decreases in other sources. If so, there may be some uncharacterized sources that lead to the relatively flat OC trend. I suggest the authors to further explore the reasons for the relative flat trend of OC.

In Fig. 2, isoprene SOC decreases by 7% from 2008 to 2016. What's the main cause for this decrease?

Minor comments

1.      Line 63. "biogenic" SOC or total SOC accounted for 3.1% of the measured OC?

2.      Line 319-323. What is the rationale to compare Beijing with Alaska?

3.      Line 324-329. Why is the concentration of biogenic SOA in Beijing is even higher than some forest sites? Higher oxidation capacity in China is one possible reason, but the sources of biogenic VOCs are also critical. Have the authors compared the concentrations or fluxes of biogenic VOCs between Beijing and forest sites?

4.      Line 438-440. In Offenberg et al., the sulfate concentration is a confounder. In other words, in Offenberg et al., it is unknown whether the enhancement in SOC is due to higher acidity or higher sulfate concentration or higher particle surface area. Have the authors investigated the relationship between apportioned SOC and sulfate concentration?

5.      Fig.4. Why are data separated into day and night in panels (a)-(d), but not in (e)-(h)?

---

## Referee Comment (RC2) · Anonymous Referee #2 · 29 Jan 2018

In the manuscript the authors apportioned the primary and secondary sources of the organic aerosols using a chemical mass balance (CMB) and trace yield methods based on 144 kinds of quantified organic species, including both primary and secondary tracers. The effectiveness of control measured on primary and secondary sources were assessed based on the obtained results. Back trajectory cluster analysis was also conducted to evaluate the influences of air mass directions on the organic aerosol sources. Environmental factors, such as temperature, O3 concentration, aerosol liquid water content, and particle acidity were also investigated to elucidate the formation mechanisms of secondary organic aerosols. The topic of the manuscript fits very well into the Atmospheric Chemistry and Physics and the manuscript is well written. Generally, I recommend the publication of the manuscript.

[Figure]

However, there are some technical details that might change the conclusion of the manuscript, which I think need to be addressed before its publication.

1. The authors spend the whole section 4.3 "Influencing factors for secondary organic formation in the summer of Beijing", discussing the factors that could influence the anthropogenic SOC (Figure 4). To get their point, they did correlation plot of the anthropogenic SOC loading with different factors and positive slope indicating enhancing effects. I found this not reasonable. What the authors really need is "multivariate analysis" or "multivariate regression". Otherwise, one factor could have influenced the behavior of the other factor and change the sign of the slope, leading to an opposite conclusion. For example $y = f(x1, x2) = x1 - 0.5 * x2$. y is positively correlated with x1, but negatively correlated with x2. You made some measurements at $x1 = 1$, $x2 = 0$ and $x1 = 2$, $x2 = 1$. The two y's you will obtain are 1 and 1.5. Then based on the authors method, one will obtain y is positively correlated with x1 (with slope of 0.5) and x2 (with slope of 0.5).

2. The authors did show in Figure 1 that the governmental control changes the Organic aerosol apportionment a bit, however, the total organic aerosol loading does not change much, or even increased (from 8.9 ug/m3 to 11.0 ug/m3) (as shown in Table 1 too). The total PM2.5 loading has decreased from 92.3 ug/m3 to 45.5 ug/m3. Then this leaves the reader wonders what have been decreased mostly? The sulfate? Nitrate? Ammonia? Or something else. The authors need to add the loading of these into Table 1. The decrease of EC from 3.3 ug/m3 to 1.8 ug/m3 is not enough to explain the more than 40 ug/m3 decrease in PM2.5.

Besides the above two comments, I also have some minor comments as listed below. 1. Line 119, by "filters" does the authors mean "quartz filter" only. Or the authors analyzed both "quartz filter" and "Teflon filter". 2. In Figure S7, are the vertical lines the measurement error bars or they indicate the daily ranges? As this could change the statement of line 270 stating that hope at PKUERS site were much higher than that of CP. 3. Line 305, the concentrations of what in CP were lower than that of PKUERS?

Please clarify.

Overall, the manuscript is well written and data presented is extensive. I recommend its publication in ACP after the aforementioned questions be addressed.

Please also note the supplement to this comment:
https://www.atmos-chem-phys-discuss.net/acp-2017-867/acp-2017-867-RC2-supplement.pdf

---

## Author Comment (AC1) · 12 Feb 2018

We thank the reviewer for careful comments and suggestions. Following is our response to the comments:

*Comments:*

*RC1:*

*This manuscript quantified 144 particulate organic species and applied chemical mass balance (CMB) model to investigate the sources of organic aerosol at two different sites (CP and PKUERS) in Beijing. The authors found that the primary sources accounted for 42.6% and 50.4% of the measured OC at CP and PKUERS, respectively, which are larger than the contribution from secondary sources. Among the secondary sources, anthropogenic VOCs contributes more to SOA than biogenic VOCs. By comparing with previous studies, the authors showed that the PM and EC concentrations have decreased. The OC concentration from many sources have decreased, with the exception of OC from gasoline engine emissions. This comparison sheds light on the evaluation of regulation policies. At last, the authors investigated the relationship between SOC concentration with temperature, ozone concentration, aerosol water content, and particle acidity. Overall, the manuscript is well-written and the data analysis is solid. I recommend publication after minor revisions to address the main comments below.*

**Major Comments**

*1. The uncertainty with tracer yield method should be better discussed. In that method, a laboratory-derived single-valued mass fraction from Kleindienst et al. (2007) is used. However, in the atmosphere, the mass fraction of molecular markers in SOA from a specific source is highly dependent on the oxidation conditions and the history of the air mass. How representative is the mass fraction value used in this study?*

We agree with the reviewer that the tracer yield method has its own limitation. The mass fraction depends on the degree of oxidation. Besides, the uncertainty also depends on the selection of the molecular tracers and the simplified procedures by using single-valued tracer mass fractions. We discuss these uncertainties in the revised manuscript. Despite of these uncertainties, tracer-yield method is confirmed to be useful in rebuilding most of the biogenic and portion of the anthropogenic SOA contributions (Offenberg et al., 2007). Previous studies showed that SOA estimated by the tracer-yield method and POA apportioned by CMB model could fully account for the OA in atmospheric atmosphere (Lewandowski et al., 2008; Kleindienst et al., 2010). Besides, researchers found that the total estimated SOC derived from tracer-yield method was in accordance with the that stemmed from EC-tracer method during summer (Ding et al., 2012; Kleindienst et al., 2010; Turpin and Huntzicker, 1995). Comparable results were also found between tracer-yield method and

positive matric factorization model (Hu et al., 2010; Zhang et al., 2009). All these results firmly demonstrated that the tracer-yield method is a valuable and convincing method to estimate the SOA contributions (X. Ding et al., 2014).

Estimations based on boundary values were generally acknowledged to have the largest source of uncertainties in the models, so those results could be used to determine the possible limit of the estimations. Also, Kleindienst et al. carried out a boundary analysis using the data from North California to measure the range of estimated SOA contributions. Results revealed that the possible contributions of isoprene, α-pinene, β-caryophyllene and toluene were within the scope of 70-130%, 50-220%, 70-120% and 60-160%, respectively. The above results were supposed to be in the acceptable range for $PM_{2.5}$ source apportionment. Besides, the standard deviations of the tracer-to-SOC ratios were suitable as a source profile uncertainty (Kleindienst et al., 2007). Despite the uncertainties above, tracer-yield represented a unique approach to estimate the SOA contributions using individual hydrocarbon precursors up to now.

The manuscript has been revised as follows (line 137-161):

"The mass fraction depends on the degree of oxidation. Besides, the uncertainty also depends on the selection of molecular tracers and the simplified procedures by using single-valued mass fractions (Yttri et al., 2011; El Haddad et al., 2011; Song et al., 2014; Guo et al. 2014b; Guo et

al., 2014c). Previous studies showed that SOA estimated by the tracer-yield method and POA apportioned by CMB model could fully account for the OA in atmospheric atmosphere (Lewandowski et al., 2008; Kleindienst et al., 2010). Besides, researchers found that the total estimated SOC derived from tracer-yield method was in accordance with the that stemmed from EC-tracer method during summer (Ding et al., 2012; Kleindienst et al., 2010; Turpin and Huntzicker, 1995). Comparable results were also found between tracer-yield method and positive matric factorization model (Hu et al., 2010; Zhang et al., 2009). All these results firmly demonstrated that the tracer-yield method is a valuable and convincing method to estimate the SOA contributions (X. Ding et al., 2014).

Estimations based on boundary values were generally acknowledged to have the largest source of uncertainties in the models, so those results could be used to determine the possible limit of the estimations. Also, Kleindienst et al. carried out a boundary analysis using the data from North California to measure the range of estimated SOA contributions. Results revealed that the possible contributions of isoprene, α-pinene, β-caryophyllene and toluene were within the scope of 70-130%, 50-220%, 70-120% and 60-160%, respectively. The above results were supposed to be in the acceptable range for PM2.5 source apportionment. Besides, the standard deviations of the tracer-to-SOC ratios were suitable as a source

profile uncertainty (Kleindienst et al., 2007). Despite the uncertainties above, tracer-yield represented a unique approach to estimate the SOA contributions using individual hydrocarbon precursors up to now."

*2. I find it very intriguing that while PM and EC concentrations have decreased from 2008 to 2016, the OC concentration is relatively constant (Table 1). As shown in Fig. 2, the contributions from many sources to OC have decreased, with the exception of vegetative detritus and gasoline engines. By eyeballing, the increases in vegetative detritus and gasoline engines seems smaller than the decreases in other sources. If so, there may be some uncharacterized sources that lead to the relatively flat OC trend. I suggest the authors to further explore the reasons for the relative flat trend of OC.*

We thank the reviewer for the comments.

We discussed detailed sources changing in the revised manuscript. The decreasing sources of the organic carbon included isoprene SOC, α-pinene SOC, toluene SOC, biomass burning, diesel exhaust and coal combustion. The increasing sources mainly contained β-caryophellene SOC, vegetative detritus, and gasoline exhausts. But the increases in β-caryophellene SOC, vegetative detritus and gasoline exhausts could not compensate for the decreases of other sources. This might be due to the unapportioned sources of OC. The uncharacterized sources may mainly

contain cooking emissions, mineral and road dust, industrial pollution, as well as other unapportioned secondary sources (Tian et al., 2016; Liu et al., 2016).

The manuscript has been revised as (line 404-412) "Compared with previous studies, except β-caryophellene SOC, vegetative detritus, and gasoline exhausts, the contributions of all other sources decreased, e.g. isoprene SOC, α-pinene SOC, toluene SOC, biomass burning, diesel exhaust, and coal combustion. However, the increases in β-caryophellene SOC, vegetative detritus and gasoline exhausts could not compensate for the decreases of other sources. This can be attributed to the larger portion of uncharacterized sources compared with 2008. The uncharacterized sources may mainly contain cooking emissions, mineral and road dust, industrial pollution as well as other uncharacterized secondary sources (Tian et al., 2016; Liu et al., 2016)."

***3. In Fig. 2, isoprene SOC decreases by 7% from 2008 to 2016. What's the main cause for this decrease?***

Thank you for the comment. In the revised manuscript we discussed the change of biogenic SOC (e.g. isoprene, α-pinene). The formation of biogenic SOA is complicated. Several factors can affect biogenic SOC formation, among which the precursor concentration is one of the crucial factor. Biogenic VOCs, i.e. isoprene, α-pinene etc. are predominantly

emitted from plant foliage in a constitutive manner. The emission rate of biogenic VOCs depends on various factors, e.g. radiation, temperature, humidity, meteorological conditions, and seasonality (Ghirardo et al., 2016). Two or more of them will act synergistically to have an effect on the concentration of biogenic SOC. Besides, the changes of the vegetation in certain area may also play a part in the change of the SOC concentration. Considering its comprehensive synergistic effect, it's difficult to determine the main reason responsible for the isoprene SOC decrease.

The manuscript has been revised as follows (line 390-401): "Compared with 2008, contributions of secondary organic aerosol decreased by 29.4%, in which biogenic SOC served as the biggest contributor to this decreasing. The formation of biogenic SOA is complicated. Several factors can affect biogenic SOC formation, among which the precursor concentration is one of the crucial factors. Biogenic VOCs, i.e. isoprene, α-pinene etc. are predominantly emitted from plant foliage in a constitutive manner. The emission rate of biogenic VOCs depends on various factors, e.g. radiation, temperature, humidity, meteorological conditions, and seasonality (Ghirardo et al., 2016). Two or more of them will act synergistically to have an effect on the concentration of isoprene SOC. Besides, the changes of the vegetation in certain area may also play a part in the change of the SOC concentration. Considering its

comprehensive synergistic effect, it's difficult to determine the main reason responsible for the isoprene SOC decrease."

*Minor comments*

1. *Line 63. "biogenic" SOC or total SOC accounted for 3.1% of the measured OC?*

The SOC here in the text means the biogenic SOC. Yang et al. (Yang et al., 2016) used tracer-yield method to estimate the biogenic secondary sources to OC during CAREBEIJING-2007. The estimated biogenic SOC included α-pinene SOC, β-caryophyllene SOC and isoprene SOC, in which isoprene-SOC was the major contributor to SOC. Therefore, the SOC in line 63 means the biogenic SOC.

We clarify this statement in the revised manuscript. The manuscript has been changed to (line 62): "Yang et al. (Yang et al., 2016) estimated the biogenic SOC to OC during CAREBEIJING-2007 field campaign, and found that the biogenic SOC accounted for 3.1% of the measured OC."

2. *Line 319-323. What is the rationale to compare Beijing with Alaska?*

Thank you for your comment. Regard to the comparison, we want to know the differences of the contributions of biogenic SOC to OC considering the quite different geographic and climate conditions to see whether different conditions would vary a great deal. Alaska is a clean

site without any anthropogenic interference, thus it's suitable to compare a relative complex anthropogenic disturbed site with a relatively clean one. Besides, restricted to the limited data of the tracer-yield method across the world, a full understanding of the SOC to OC with different background seems necessary for better understanding of the contributions of secondary organic aerosol with different origins.

3. *Line 324-329. Why is the concentration of biogenic SOA in Beijing is even higher than some forest sites? Higher oxidation capacity in China is one possible reason, but the sources of biogenic VOCs are also critical. Have the authors compared the concentrations or fluxes of biogenic VOCs between Beijing and forest sites?*

Thank you the reviewer for the suggestion. We discussed the reason for higher biogenic SOA concentration in Beijing. Firstly, the complex oxidation conditions. Higher oxidation capacity in China may fasten the chemical lifetime of reactive gases and accelerate the aerosol aging process which thus leads to an increase in biogenic SOA (Ghirardo et al., 2016). Secondly, with complicated emissions of anthropogenic VOCs, the anthropogenic emissions can also lead to an enhancement of biogenic SOA (Hoyle et al., 2011). We also compare the isoprene concentration according to some literature. Wang et al. (Wang et al., 2010) discovered that the mean isoprene concentration was 0.24 ppbv at PKUERS in June,

2008. Lappalainen et al. (Lappalainen et al., 2009) measured the isoprene concentration of the boreal forest in Hyytiala and found that the mean concentration of isoprene was 0.15 ppbv. Therefore, at least for isoprene, the concentration in China is much higher than that of the forest site.

The text has been revised as following (line 349-360):

"Higher oxidation capacity in China may fasten the chemical lifetime of reactive gases and accelerate the aerosol aging process which leads to an increase in biogenic SOA (Ghirardo et al., 2016). Another possible reason derived from the complicated emissions of anthropogenic VOCs which can lead to an enhancement of secondary organic aerosol formation from biogenic precursors (Hoyle et al., 2011) We also compare the isoprene concentration with the forest site according to some literatures. Wang et al. (Wang et al., 2010) discovered that the mean isoprene concentration was 0.24 ppbv at PKUERS in June 2008. Lappalainen et al. (Lappalainen et al., 2009) measured the isoprene concentration of the boreal forest in Hyytiala and found that the mean concentration of isoprene was 0.15 ppbv. This comparable, or even higher concentration of isoprene may be due to different sources of biogenic VOCs."

*4. Line 438-440. In Offenberg et al., the sulfate concentration is a confounder. In other words, in Offenberg et al., it is unknown whether the enhancement in SOC is due to higher acidity or higher sulfate concentration or higher particle surface area. Have the authors*

*investigated the relationship between apportioned SOC and sulfate concentration?*

We thank the reviewer for the kind suggestion.

According to your suggestion, we did the analysis to investigate the relationship between apportioned SOC and sulfate concentration. The results are shown in the figures below. The apportioned SOC was positively correlated with the concentration of sulfate. The correlation coefficient $R^2$ were 0.41 and 0.45 for CP and PKUERS, respectively, indicating that the increase of SOC may be influenced by the sulfate aerosol concentration. As such, the increase in the SOC concentration is likely arise from the acid-catalyzed reactions with the participation of sulfate aerosols.

Two figures have been added into Fig. 4 (i)(j). Explanation of Fig.4 (i)(j) has been added into the text as (line 495-500) "We also analyzed the relationship between apportioned SOC and sulfate concentration and found that the apportioned SOC increased with the increase of sulfate concentration. The correlation coefficient $R^2$ were 0.41 and 0.45 for CP and PKUERS, respectively, indicating that the increase of SOC may be influenced by the sulfate aerosol. As such, the increase in SOC is likely arise from the acid-catalyzed reactions with the participation of sulfate aerosols."

[Figure]

[Figure]

**5. Fig.4. Why are data separated into day and night in panels (a)-(d), but not in (e)-(h)?**

Thank you for your comment.

We found that the correlations between SOC and O$_3$/temperature are different for daytime and nighttime samples. However, it's not significant for water content and H$^+$. We add some description in the revised text.

The manuscript has been revised by adding the following contents (line 439-443):

[revised manuscript text omitted]

---

## Author Comment (AC2)

**Response to reviewer 2**

We thank the reviewer for careful comments and suggestions. Following is our response to the comments:

*Comments RC2:*

*In the manuscript the authors apportioned the primary and secondary sources of the organic aerosols using a chemical mass balance (CMB) and trace yield methods based on 144 kinds of quantified organic species, including both primary and secondary tracers. The effectiveness of control measured on primary and secondary sources were assessed based on the obtained results. Back trajectory cluster analysis was also conducted to evaluate the influences of air mass directions on the organic aerosol sources. Environmental factors, such as temperature, O3 concentration, aerosol liquid water content, and particle acidity were also investigated to elucidate the formation mechanisms of secondary organic aerosols. The topic of the manuscript fits very well into the Atmospheric Chemistry and Physics and the manuscript is well written. Generally, I recommend the publication of the manuscript.*

*However, there are some technical details that might change the conclusion of the manuscript, which I think need to be addressed before its publication.*

Main Comments

**1. The authors spend the whole section 4.3 "Influencing factors for secondary organic formation in the summer of Beijing", discussing the factors that could influence the anthropogenic SOC (Figure 4). To get their point, they did correlation plot of the anthropogenic SOC loading with different factors and positive slope indicating enhancing effects. I found this not reasonable. What the authors really need is "multivariate analysis" or "multivariate regression". Otherwise, one factor could have influenced the behavior of the other factor and change the sign of the slope, leading to an opposite conclusion. For example $y = f(x1, x2) = x1 - 0.5 * x2$. y is positively correlated with x1, but negatively correlated with x2. You made some measurements at $x1 = 1, x2 = 0$ and $x1 = 2, x2 = 1$. The two y's you will obtain are 1 and 1.5. Then based on the authors method, one will obtain y is positively correlated with x1 (with slope of 0.5) and x2 (with slope of 0.5).**

We thank the reviewer for the comments.

What we want to do in section 4.3 is to roughly discuss the influencing factors that can have an impact on the SOC concentration, thus shed light on further study to concentrate on the influencing factors concerning the SOA formation. So we use univariate analysis to see which factor may influence the apportioned SOC and see the correlation

between the potential influencing factors and the apportioned SOC. The correlation between different parameters could at least enlighten us of the influencing factors for SOA formation in megacities such as Beijing under the complex air pollution conditions.

Besides, we did the multivariate analysis as the reviewer suggested. The multiple regression analysis was used to investigate the relationship between SOC and water content, $H^+$, temperature, and ozone concentration. The multiple regression equation was as following:

SOC=0.5495 + 0.052×water content + 5.24×$H^+$ + 0.01085×temp + 0.01054×$O_3$

The correlation coefficient R=0.73. All the influencing factors have positive impact on the SOC concentration. According to our results, $H^+$ concentration has significantly great impact on SOC formation.

Anyway, all these influencing factors can interact with each other. Therefore, the multivariate analysis also has large uncertainties.

*2. The authors did show in Figure 1 that the governmental control changes the Organic aerosol apportionment a bit, however, the total organic aerosol loading does not change much, or even increased (from 8.9 ug/m³ to 11.0 ug/m³) (as shown in Table 1 too). The total PM$_{2.5}$ loading has decreased from 92.3 ug/m³ to 45.5ug/m³. Then this leaves the reader wonders what have been decreased mostly? The sulfate?*

*Nitrate? Ammonia? Or something else. The authors need to add the loading of these into Table 1. The decrease of EC from 3.3 ug/m³ to 1.8 ug/m³ is not enough to explain the more than 40 ug/m³ decrease in PM$_{2.5}$.*

We agree with the reviewer. Additional discussion about other compounds, i.e. inorganic components, was included in the revised text.

We could see from table 1 that after the government took control strategies, the concentrations of PM$_{2.5}$, EC decreased significantly since 2008. However, the OC concentrations didn't show the same tendency with PM$_{2.5}$ and EC. To elucidate the reasons for the dramatic decrease of PM$_{2.5}$, we compared the data of the main inorganic water soluble ions i.e. sulfate, nitrate and ammonia (relevant data has been added to table 1). Results showed that the averaged concentration of water inorganic water soluble ions decreased from 2008, with sulfate decreased from 35.6 μg/m³ to 4.7 μg/m3, nitrate decreased from 7.9 μg/m³ to 2.4 μg/m³, ammonia decreased from 15.2 μg/m³ to 3.5 μg/m³. The significant decrease of SNA and EC confirmed the effectiveness of the drastic measures taken by the government. Therefore, the reduction of fine particulate matter was mainly due to the well controlling of the EC and inorganic particulate matter such as sulfate, nitrate and ammonia, especially the dramatic decrease of sulfate (86.8% from 2008 to 2016).

The relevant data of SNA has been added to table 1, and the discussion

about the decrease of $PM_{2.5}$ was as follows (line 222-228) "Relevant data of main WSICs (sulfate, nitrate and ammonia) during 2008 to 2016 were also included in table 1 to better elucidate the drastic decrease of fine particulate matter in recent years. Results showed that the daily average concentration of WSICs decreased from 2008 to 2016, with sulfate decreased from 35.6 μg/m$^3$ to 4.7 μg/m$^3$, nitrate decreased from 7.9 μg/m$^3$ to 2.4 μg/m3, ammonia decreased from 15.2 μg/m$^3$ to 3.5 μg/m$^3$. The significant decrease of WSICs confirmed the effectiveness of the control measures taken by the government" and "Therefore, we could draw a conclusion that the drastic decrease of fine particulate matter in Beijing was mainly due to the well-controlled EC and WSICs, with negligible contribution of OC".

*Besides the above two comments, I also have some minor comments as listed below.*

*1. Line 119, by "filters" does the authors mean 〝quartz filter〞 only. Or the authors analyzed both 〝quartz filter〞 and 〝Teflon filter〞.*

Thank you for your comment.

The "filters" mentioned here was referred to quartz filters only. As is mentioned above, the four-channel samplers (TH-16A, Tianhong, China) consisted of three quartz filer channel and one Teflon filter channel.

Teflon filter was weighed and used to calculate the concentration of $PM_{2.5}$ and analyze the water-soluble inorganic compounds. The quartz filters were used to analyze the EC, OC and the particulate organic matters. Here, the "filters" referred politically to quartz filters.

The manuscript has been altered from "The filters were then ultrasonically extracted with methanol: dichloromethane (v:v=1:3) solvent in water bath (temperature < 30 $^o$C) for 3 times" to "The quartz filters were then ultrasonically extracted with methanol: dichloromethane (v:v=1:3) solvent in water bath (temperature < 30 $^o$C) for 3 times" to avoid ambiguity (line 110).

2. *In Figure S7, are the vertical lines the measurement error bars or they indicate the daily ranges? As this could change the statement of line 270 stating that hope at PKUERS site were much higher than that of CP.*

We thank the reviewer for the comment.

The vertical lines represent the standard deviation of the daily concentrations. For comparison, we compared the daily average values for simplification and thus stated that the hopanes at the urban site PKUERS were higher than that of CP.

The relevant context "For all the species, the histogram showed the average daily concentrations with error bars representing the standard

deviations" has been added to the manuscript (line 256-257)

**3. Line 305, the concentrations of what in CP were lower than that of PKUERS?**

Thank you for your comment.

It's the concentration of 2,3-dihydroxy-4-oxopentanoic acid that was lower in CP compared with PKUERS. We revise this sentence to make it clear: "However, the 2,3-dihydroxy-4-oxopentanoic acid concentrations in CP were lower than that of PKUERS..." (line 318-319)

---

## Referee Report (RR1)

Review comments on "Primary and secondary organic aerosols in 2016 summer of Beijing" by Tang et al

The authors have satisfactorily addressed all my concerns. I suggest the publication of the manuscript in ACP.